# Beyond Boundaries: A Novel Data-Augmentation Discourse for Open Domain Generalization

**Shirsha Bose** *shirshabosecs@gmail.com*
*Technical University of Munich*

**Ankit Jha** *ankitjha16@gmail.com*
*Indian Institute of Technology Bombay*

**Hitesh Kandala** *khitesh2000@gmail.com*
*Indian Institute of Technology Bombay*

**Biplab Banerjee** *getbiplab@gmail.com*
*Indian Institute of Technology Bombay*

**Reviewed on OpenReview:** *https://openreview.net/forum?id=jpZmhiIys1*

## Abstract

The problem of Open Domain Generalization (ODG) is multifaceted, encompassing shifts in domains and labels across all source and target domains. Existing approaches have encountered challenges such as style bias towards training domains, insufficient feature-space disentanglement to highlight semantic features, and discriminativeness of the latent space. Additionally, they rely on a confidence-based target outlier detection approach, which can lead to misclassifications when target open samples visually align with the source domain data. In response to these challenges, we present a solution named ODG-NET. We aim to create a direct open-set classifier within a *discriminative*, *unbiased*, and *disentangled* semantic embedding space. To enrich data density and diversity, we introduce a generative augmentation framework that produces *style-interpolated* novel domains for closed-set images and novel pseudo-open images by *interpolating the contents of paired training images*. Our augmentation strategy skillfully utilizes *disentangled style and content information* to synthesize images effectively. Furthermore, we tackle the issue of style bias by representing all images in relation to all source domain properties, which effectively accentuates complementary visual features. Consequently, we train a multi-class semantic object classifier, incorporating both closed and open class classification capabilities, along with a style classifier to identify style primitives. The joint use of style and semantic classifiers facilitates the disentanglement of the latent space, thereby enhancing the generalization performance of the semantic classifier. To ensure discriminativeness in both closed and open spaces, we optimize the semantic feature space using novel metric losses. The experimental results on six benchmark datasets convincingly demonstrate that ODG-NET surpasses the state-of-the-art by an impressive margin of $1 - 4\%$ in both open and closed-set DG scenarios.

## 1 Introduction

Domain Generalization (DG), as explored in the work of Zhou et al. (2022), aims to establish a shared embedding space derived from labeled source domains, which can then be applied to an unseen target domain. However, current DG methods are primarily tailored to closed-set scenarios, such as those seen in Zhou et al. (2020b) and Zhou et al. (2020a), where both source and target domains possess identical label sets. Nonetheless, this approach might not always be viable in dynamic real-world contexts, as exemplified by Robotics, where a navigating robot might encounter categories that are either common or unique to its surroundings Zhao & Shen (2022). This realization underscores the necessity of tackling the more pragmatic and intricate realm of Open Domain Generalization (ODG) Shu et al. (2021), which revolves around training on labeled source domains housing both shared and domain-specific categories.

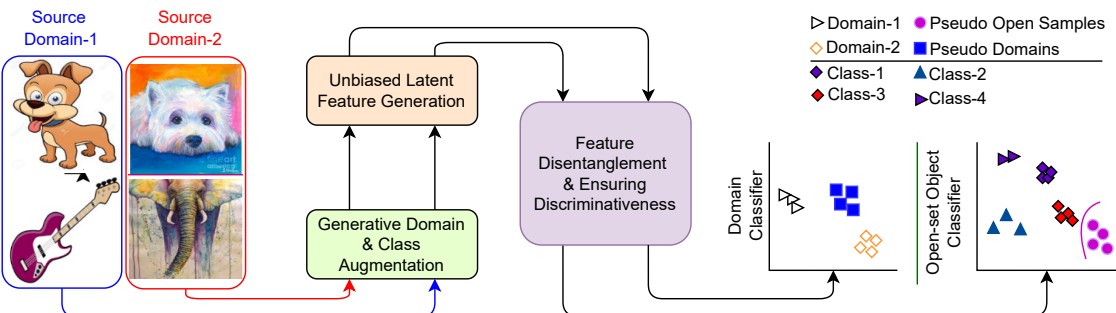

Figure 1: The working principle of ODG-NET. Given a number of source domains with shared and private classes for each domain, our algorithm follows three simple stages to obtain an effective semantic open-set object classifier.

In the context of ODG, the target domain consists of samples belonging to either familiar classes or novel classes exclusive to that particular domain, thereby introducing several noteworthy challenges. Firstly, a substantial data imbalance arises due to the uneven representation of known classes within the source domains. Secondly, the task of establishing an embedding space that is both domain-agnostic and discriminatory becomes formidable in light of unregulated shifts in both domains and labels. Lastly, the absence of prior knowledge concerning the open space in the target domain adds to the complexity.

While one potential approach to addressing ODG could involve the fusion of a pre-existing closed-set DG technique with a readily available open-set recognition (OSR) method, such as Openmax Bendale & Boult (2016b), this strategy may not yield optimal results. The DG technique could potentially be influenced by the domain-specific classes found in ODG, which are subject to significant under-representation.

Surprisingly, Open Domain Generalization (ODG) has garnered little attention in the Domain Generalization (DG) literature, with DAML Shu et al. (2021) being the sole model explicitly designed for ODG. However, our extensive investigation revealed three significant limitations in DAML's approach. Initially, DAML augments source domains via multi-domain mix-up features, using a Dirichlet-distribution based weight scheme, where identical weight configurations correspond to labels for generated features. However, merging content and style details in the raw latent features might result in semantically inconsistent feature-label pairs. This was evidenced in a PACS dataset Li et al. (2017) experiment, where classifying the synthesized features yielded notably low accuracy. Additionally, DAML overlooks disentangling content features from domain attributes, potentially introducing biases towards specific styles, hindering adaptability across diverse visual domains. Furthermore, DAML's outlier rejection relies on thresholding source domain classifier responses, that can produce erroneous results, especially when target open samples visually align with certain source domain classes. These limitations underscore the imperative need for further exploration and refinement of ODG techniques, thereby unlocking their full potential and meaningfully addressing the inherent complexities.

To overcome these challenges, we introduce an innovative approach for training a robust generalized semantic open-set classifier. This classifier can distinguish known and potential new samples in an unbiased and disentangled embedding space. However, we face a hurdle as we lack representative open-space samples during training. To address this, we propose a novel technique called "sample hallucination," which generates these essential samples, facilitating comprehensive classifier training. Additionally, we tackle the issue of class imbalance in ODG by diversifying the appearances of training classes. Our method emphasizes the importance of representing images in a feature space unaffected by training domains. To achieve this, we advocate separating semantic features from style elements, revealing each image's true content. Ultimately, we ensure that the semantic space effectively discriminates between classes, enabling our classifier to make precise distinctions across various categories. These innovations pave the way for more effective and accurate open domain generalization.

**Our proposed ODG-NET**: In this paper, we introduce our novel model, ODG-NET (depicted in Fig. 1), which addresses the aforementioned concerns comprehensively. ODG-NET consists of three pivotal modules, each devoted to addressing model bias, feature disentanglement, and the creation of a discriminative semantic feature space.

At the core of our research is the aim to enrich the existing source domains using two types of synthesized images generated by a novel conditional GAN. These image types serve specific purposes: expanding the diversity of closed-set classes and creating representative pseudo-open images. The first image type, called domain or style mix-up images, involves a sophisticated interpolation of style properties from source domains using a Dirichlet distribution. This

approach introduces new domains, diversifying the style of source images while preserving their inherent semantic object characteristics. The second type, known as pseudo open-space images, results from skillful interpolation of both domain and class identifiers from source domain images, achieved through Dirichlet sampling. To ensure the creation of diverse and meaningful samples, we've introduced diversification regularization to prevent potential mode collapse during the generation of domain or label-interpolated samples. Additionally, a structural cycle consistency mechanism has been implemented to maintain the structural integrity of the generated images.

Our approach tackles a range of formidable challenges, spanning from class imbalance and limited style diversity to the absence of an open-space prior. A unique aspect of our methodology lies in its capacity to unify label and domain mix-up concepts while offering customization in conditioning. This surpasses existing augmentation methods, which are limited to style or random image mix-ups Mancini et al. (2020a); Zhou et al. (2021). ODG-NET strives for an unbiased latent embedding space, devoid of source domain bias. We achieve this through an innovative approach involving the training of domain-specific classifiers, which adeptly capture domain-specific features from each source domain. Consequently, each image is represented as a concatenation of features from all domain-specific models, creating a comprehensive and impartial embedding.

*To disentangle domain-specific attributes from semantic object features in latent representations, we train two attention-driven classifiers: a domain classifier for domain label identification and an object classifier for recognizing class labels from the augmented domain set. This enriches our model's grasp of object semantics while significantly mitigating the influence of domain-specific artifacts.* For a highly discriminative semantic feature space, we introduce a contrastive loss among known classes, accentuating differences between various categories. Additionally, our entropy minimization objective strategically pushes pseudo outliers away from the known-class boundary, bolstering the model's robustness.

We summarize our **major contributions** as:

[-] In this paper, we introduce ODG-NET, an end-to-end network that tackles the challenging ODG problem by jointly considering closed and open space domain augmentation, feature disentanglement, and semantic feature-space optimization.

[-] To synthesize augmented images that are diverse from the source domains, we propose a novel conditional GAN with a cycle consistency constraint and an anti-mode-collapse regularizer that interpolates domain and category labels. We also adopt a classification-based approach for feature disentanglement. Finally, we ensure the separability of the semantic feature space for closed and open classes through novel metric objectives.

[-] We evaluate ODG-NET on six benchmark datasets in both open and closed DG settings. Our experiments demonstrate that ODG-NET consistently outperforms the literature. For instance, on ODG for Multi-dataset Shu et al. (2021) and on closed DG for DomainNet Peng et al. (2019), ODG-NET outperforms the previous state-of-the-art by approximately $3\%$.

## 2 Related Works

**Open-set learning and open-set domain adaptation:** The Open-set Recognition (OSR) Bendale & Boult (2016c); Kong & Ramanan (2021); Pal et al. (2023b); Vaze et al. (2021) challenge involves effectively identifying novel, unknown-class samples during testing, leveraging training samples from known closed-set classes. However, OSR doesn't account for any differences in distributions between the training and test sets. Another relevant problem is Open-set Domain Adaptation (OSDA) Panareda Busto & Gall (2017); Saito et al. (2018); Kundu et al. (2020); Bucci et al. (2020), which addresses the scenario of a labeled source domain and an unlabeled target domain. The target domain contains unlabeled samples from the same semantic classes as the source domain, along with novel-class samples unique to the target domain. OSDA operates in a transductive manner, where both source and target domains are simultaneously employed during training. In contrast, Open Domain Generalization (ODG) sets itself apart from OSR and OSDA. In ODG, the target domain remains unseen during training, making it distinct. Additionally, the multiple source domains consist of a combination of shared and private categories. This diversity of categories, including shared and domain-specific ones, renders ODG even more intricate than the other tasks.

**(Open) DG:** DG refers to the problem of learning a supervised learning model that is generalizable across any target distribution without any prior knowledge. The initial studies in closed-set DG focused on domain adaptation (DA) Li

et al. (2020); Wang et al. (2021); Li et al. (2021a) due to the disparity in domain distributions. Several DG methods have since been developed, such as self-supervised learning Carlucci et al. (2019), ensemble learning Xu et al. (2014), and meta-learning Patricia & Caputo (2014); Wang et al. (2020b); Li et al. (2019b; 2018a; 2019a); Huang et al. (2020). To address the domain disparity, the concept of domain augmentation Li et al. (2021c); Kang et al. (2022); Zhou et al. (2020b; 2021); Zhang et al. (2022) was introduced, which involves generating pseudo-domains and adding them to the available pool of domains. Subsequently, the notion of ODG was introduced in Shu et al. (2021), which is based on domain-augmented meta-learning. To solve the single-source ODG problem, Zhu & Li (2021) and Yang et al. (2022) further extended the idea of multi-source ODG. See Zhou et al. (2022) for more discussions on DG.

*Our proposed* ODG-NET *represents a significant departure from DAML Shu et al. (2021). Unlike their ad-hoc feature-level mix-up strategy, we introduce a more robust augmentation technique that leverages generative modeling to seamlessly synthesize pseudo-open and closed-set image samples. Additionally, we take a direct approach to learning an open-set classifier in a meaningful and optimized semantic space, in contrast to the source classifier's confidence-driven inference used in Shu et al. (2021). As a result,* ODG-NET *is better suited to handling open samples of different granularities.*

**Augmentation in DG:** Data augmentation is a crucial technique in DG, and it can be implemented using various methods such as variational autoencoders, GANs, and mixing strategies Goodfellow et al. (2020); Kingma & Welling (2013); Zhang et al. (2017). For instance, Rehman *et al.* Rahman et al. (2019) used ComboGAN to generate new data and optimized ad hoc domain divergence measures to learn a domain-generic space. Zhou *et al.* Zhou et al. (2020b) combined GAN-based image generation with optimal transport to synthesize images different from the source data. Gong *et al.* Gong et al. (2019) treated generation as an image-to-image translation process and extracted intermediate images given an image pair. Similarly, Li et al. (2021b) used adversarial training to generate domains instead of samples. Mix-up, on the other hand, generates new data by interpolating between a pair of samples and their labels. Recently, mix-up techniques Yun et al. (2019); Mancini et al. (2020a); Zhou et al. (2021) have become popular in the DG literature, applied to either the image or feature space.

The augmentation approach used by ODG-NET stands out from the existing literature by going beyond simple style or image mix-up. Our approach ensures that the object properties of images remain intact when using style mix-up, and we also have control over label mix-up to generate pseudo-open samples that can scatter the open space with varying levels of similarity to the source domains.

*Among the existing augmentation strategies, Zhou et al. (2020b) and Gong et al. (2019) are the closest to our approach as they both use conditional GANs. However, there are several key differences between our method and theirs: (a) Gong et al. (2019) requires paired training data to sample intermediate pseudo-stylized images, whereas we use conditional generation without the need for paired data; (b) Zhou et al. (2020b) uses extrapolation for domains, which is ill-posed, while we use Dirichlet distributions to interpolate domains and classes; and (c) while both Zhou et al. (2020b) and Gong et al. (2019) use style mix-up for closed-set data, we generate both closed and pseudo-open samples judiciously.*

**Disentangled representation learning**. Disentangled feature learning refers to the process of modeling distinct and explanatory data variation factors. As per Dittadi et al. (2020), disentanglement can aid in out-of-distribution tasks. Previous efforts have focused on disentangling semantic and style latent variables in the original feature space using encoder-decoder models Wang et al. (2022); Cai et al. (2019), causality Ouyang et al. (2021), or in the Fourier space Wang et al. (2022). These models are complex and require sophisticated knowledge to improvement the feature learning of the models. *In contrast,* ODG-NET *proposes to use simple to implement yet effective, attention-based classifiers, to separate the style and semantic primitives from the latent visual representations.*

## 3 Problem Definition and Proposed Methodology

In the context of ODG, we have access to multiple source domains denoted as $\mathcal{D} = \{\mathcal{D}_1, \mathcal{D}_2, \cdots, \mathcal{D}_{\mathcal{S}}\}$. Each of these domains has different distributions and contains a combination of domain-specific and shared categories. During training, we use labeled samples from each domain $\mathcal{D}_s = (x_s^i, y_s^i)_{i=1}^{n_s}$, where $y_s \in \mathcal{Y}_s$ is the label for $x_s \in \mathcal{X}_s$. The total number of classes in $\mathcal{D}$ is denoted by $\mathcal{C}$. The target domain $\mathcal{D}_{\mathcal{T}} = \{x_t^j\}_{j=1}^{n_t}$ has a distribution that is different from that of $\mathcal{D}$. It consists of unlabeled samples that belong to one of the source classes present in $\mathcal{D}$ or novel classes

that were not seen during training. The objective is to model a common classifier that can reject outliers while properly recognizing the known class samples given $\mathcal{D}$ and then evaluate its performance on $\mathcal{D}_\mathcal{T}$.

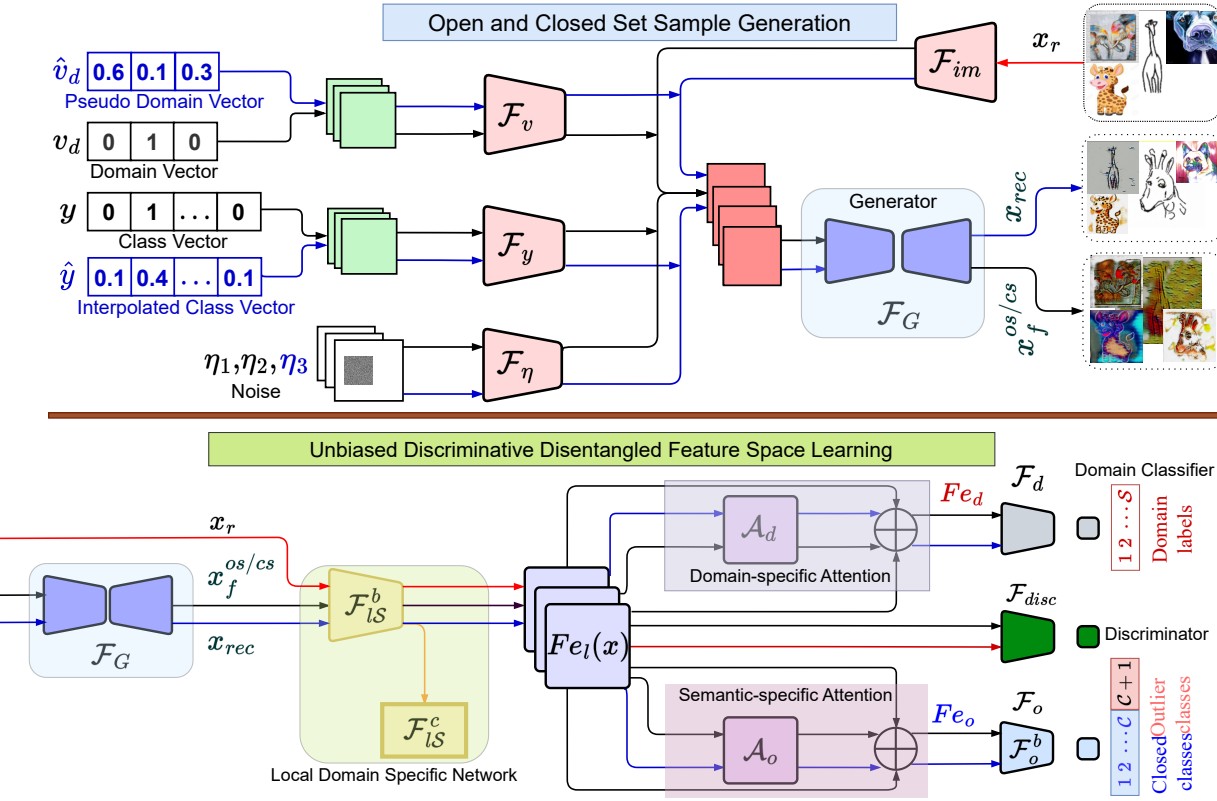

Figure 2: A depiction of the ODG-NET architecture. It shows the model components: the embedding networks: $(\mathcal{F}_{im}, \mathcal{F}_v, \mathcal{F}_y , \mathcal{F}_\eta)$, cGAN consisting of $(\mathcal{F}_G, \mathcal{F}_{disc})$, the local domain-specific classifiers $\{\mathcal{F}_l^s = (\mathcal{F}_{ls}^b, \mathcal{F}_{ls}^c)\}_{s=1}^{\mathcal{S}}$, and the global domain and semantic classifiers $(\mathcal{F}_d, \mathcal{F}_o)$ with corresponding attention blocks $\mathcal{A}_d$ and $\mathcal{A}_o$, respectively. Colors indicate the flow of information for different data items.

In our formulation, each domain/style in $\mathcal{D}$ is represented using an $\mathcal{S}$-dimensional one-hot vector $v_d$. In contrast, a pseudo-domain (synthesized style) is represented by $\hat{v}_d$, which is sampled from a Dirichlet distribution with parameter $\alpha$ and has the same length as $v_d$. For instance, if $\mathcal{S} = 3$, a source domain can be represented as a three-dimensional one-hot vector (e.g., $[0, 0, 1]$), while a $\hat{v}_d$ could be $[0.2, 0.3, 0.5]$.

Similarly, we denote the label $y$ as a $\mathcal{C}$-dimensional one-hot vector. On the other hand, an interpolated label space is represented by $\hat{y}$, which is sampled in the same way as $\hat{v}_d$. A real image-label pair from $\mathcal{D}$ is denoted as $(x_r, y)$. In contrast, a cGAN synthesized image is denoted by $(x_f^{cs}, y)$ or $(x_f^{os}, \hat{y})$ depending on whether it represents a style-interpolated closed-set image with label $y$ or a joint style and label interpolated pseudo-open image with label $\hat{y}$, respectively. Finally, to aid in open-set classification, we introduce a label space $\tilde{y} \in \mathbb{R}^{\mathcal{C}+1}$. The first $\mathcal{C}$ indices are reserved for closed-class samples, while the $\mathcal{C} + 1$-th index is used for pseudo-outlier samples.

## 3.1 Architecture and training overview for ODG-NET

Our objective is to craft a direct open-set classifier operating within a disentangled, discriminative, and unbiased semantic feature space. Additionally, we introduce a method for generating training samples for the open space by proposing the creation of pseudo-open samples through a generative module.

To accomplish our aims, we introduce ODG-NET, composed of four modules (illustrated in Fig. 2). First and foremost, ODG-NET employs a generative augmentation strategy, utilizing a **conditional GAN** equipped with a U-Net-based generator denoted as $\mathcal{F}_G$ and a binary discriminator referred to as $\mathcal{F}_{disc}$. We condition $\mathcal{F}_G$ on four variables: the domain label $v_d/\hat{v}_d$, the class label $y/\hat{y}$, an input image $x_r/x_f^{cs}/x_f^{os}$, and a noise tensor $\eta_1/\eta_2/\eta_3$

sampled from predefined distributions. To ensure the proper combination of these conditioning variables, we propose the use of separate embedding networks $(\mathcal{F}_{im}, \mathcal{F}_v, \mathcal{F}_y, \mathcal{F}_\eta)$ to encode the image, domain label, class label, and noise into meaningful latent representations. We train $\mathcal{F}_G$ to generate two types of images: (i) $(x_f^{cs}, y)$ when conditioned on $(x_r, \hat{v}_d, y, \eta_1)$, where $x_f^{cs}$ retains the class label $y$ of $x_r$ while the style changes according to $\hat{v}_d$, and (ii) $(x_f^{os}, \hat{y})$ when conditioned on $(x_r, v_d/\hat{v}_d, \hat{y}, \eta_2)$, thereby modifying the semantic and stylistic characteristics of $x_r$ according to $\hat{y}$ and $v_d/\hat{v}_d$ in $x_f^{os}$. We employ a standard min-max formulation to train the conditional GAN and introduce a regularizer to ensure that the generated samples do not closely resemble the data from $\mathcal{D}$ (as expressed in Eq. 1). Furthermore, we introduce a cycle consistency loss to maintain the semantic consistency of the generated images (as indicated in Eq. 2). To achieve an unbiased latent feature space, we propose representing all images with respect to the feature space of all source domains. We introduce $\mathcal{S}$ **local source-domain specific networks**, which comprise a feature backbone and a classification module, denoted as $\mathcal{F}_l^s = (\mathcal{F}_{ls}^b, \mathcal{F}_{ls}^c)$ and are trained on $\mathcal{D}_s$. We aggregate feature responses from all $\mathcal{F}_{ls}^b$ to obtain the latent representation $Fe_l(x) = [\mathcal{F}_{l1}^b; \mathcal{F}_{l2}^b; \cdots; \mathcal{F}_{lS}^b]$ for a given image $x$ (as described in Eq. 3). To disentangle domain-dependent properties from semantic object features within $Fe_l(x)$, we introduce **global domain and semantic object classifiers**, denoted as $\mathcal{F}_d$ with $\mathcal{S}$ output nodes and $\mathcal{F}_o$ with $\mathcal{C} + 1$ output nodes (expressed in Eq. 4-5), which are shared across domains. We employ spectral-spatial self-attention modules $\mathcal{A}_d$ and $\mathcal{A}_o$ to highlight domain and semantic object features from $Fe_l$, resulting in $Fe_d$ and $Fe_o$. We aim to ensure the discriminative quality of the semantic embedding space (outputs of the feature encoder of $\mathcal{F}_o$, denoted as $\mathcal{F}_o^b$) through novel metric losses, which encourage the separation of all closed and pseudo-open class samples (as indicated in Eq. 6-7). In the following sections, we delve into the details of the proposed loss functions.

### 3.2   Loss functions, training, and inference

**Regularized cGAN objectives with structural cycle consistency for image synthesis:** As previously mentioned, we employ a cGAN model to generate potential open-space images. Within this framework, $\mathcal{F}_G$ and $\mathcal{F}_{disc}$ engage in a min-max adversarial game. The primary objective of $\mathcal{F}_{disc}$ is to accurately discern between real and synthesized images, while $\mathcal{F}_G$ endeavors to deceive $\mathcal{F}_{disc}$. To prevent the generated images from being too similar to those in $\mathcal{D}$, we introduce a regularization term denoted as $\mathcal{W}$. This regularization term penalizes scenarios in which the real and synthesized images, represented by their respective features $Fe_l(x_r)$ and $Fe_l(x_f^{cs/os})$, become indistinguishable for slightly different values of $(v_d, \hat{v}_d)$ or $(y_d, \hat{y}_d)$. Here, $\delta$ represents the cosine similarity, and $\epsilon$ is a small constant. Essentially, even if $\delta(v_d, \hat{v}_d)$ or $\delta(y_d, \hat{y}_d)$ tends toward 1, $\mathcal{W}$ enforces $\delta(Fe_l(x_r), Fe_l(x_f^{cs/os}))$ to approach 0, minimizing the loss. The comprehensive loss formulation is presented below.

$$\mathbf{L_{Gan}} = \underset{P_{\mathcal{D}}, P_{noise}^{os/cs}}{\mathbb{E}} \Big[ log(\mathcal{F}_{disc}(Fe_l(x_r))) + log(1 - \mathcal{F}_{disc}(Fe_l(x_f^{cs/os}))) + \beta \underbrace{\frac{\delta(Fe_l(x_r), Fe_l(x_f^{os/cs})) + \epsilon}{\delta(v_d, \hat{v}_d) + \delta(y_d, \hat{y}_d) + \epsilon}}_{\mathcal{W}} \Big] \tag{1}$$

In this context, $P_{\mathcal{D}}$, $P_{noise}^{cs}$, and $P_{noise}^{os}$ denote the data distribution of the source domains in $\mathcal{D}$ and the noise used to generate closed-set and open-set samples, respectively. We set $P_{noise}^{cs} = \mathcal{N}(0, \mathbb{I})$, and $P_{noise}^{os} = \mathcal{N}(0, \sigma)$, where $\sigma$ is a large value. Our goal is to limit the space of generated closed-set images so that they represent similar semantic concepts while allowing for more scattering in the pseudo-open space, aiding in learning a robust open-set classifier.

To maintain the structural robustness of $\mathcal{F}_G$ against variations in style or label, we propose a method for reconstructing $x_r$. We take into account the embeddings of the synthesized $x_f^{cs/os}$, the actual class label $y$, the domain identifier $v_d$, and a noise vector $\eta_3 \in \mathcal{N}(0, \mathbb{I})$ as inputs to $\mathcal{F}_G$: $x_r^{rec} = \mathcal{F}_G(x_f^{os/cs}, v_d, y, \eta_3)$. By following the path $x_r \rightarrow x_f^{cs/os} \rightarrow x_r^{rec}$, we ensure that $x_f^{cs/os}$ represents a meaningful image rather than noisy data.

To compute the loss, we use the standard $\ell_1$ distance between the original real domain images and the remapped images $x_r^{rec}$, given by:

$$\mathbf{L_{rec}} = \underset{P_{\mathcal{D}}, \mathcal{N}(0, \mathbb{I})}{\mathbb{E}} [||x_r - x_r^{rec}||_1^1]. \tag{2}$$

**Learning style agnostic latent representations for all the images:** Subsequently, our aim is to guarantee that the latent feature embeddings of images remain impartial and not skewed toward any particular training source domain. To mitigate the risk of overfitting to any specific source domain, we introduce a method wherein input images are represented based on the characteristics of all source domains, leveraging the feature representation $Fe_l(x)$. This approach constructs a multi-view representation space that encompasses diverse and complementary perspectives of the images.

To achieve this, we train $\mathcal{F}_l^s$ using $\mathcal{D}_s$ where $s$ belongs to $\{1, 2, \cdots, \mathcal{S}\}$. We consider $\mathcal{S}$ multiclass cross-entropy losses ($\mathbf{L_{CE}}$) for this purpose (as shown in Eq. 3), where $P_{\mathcal{D}}^s$ represents the data distribution for the $s^{th}$ source domain.

$$\mathbf{L_{local}} = \frac{1}{\mathcal{S}} \sum_{s \in \{1, 2, \cdots, \mathcal{S}\}} \mathbb{E}_{P_{\mathcal{D}}^s} [\mathbf{L_{CE}}(\mathcal{F}_{ls}^c(x_s), y_s)]. \tag{3}$$

**Disentangling latent features of the images to highlight the semantic contents through global classifiers:** Simultaneously, our objective is to disentangle domain-specific attributes from the semantic object features within the previously derived latent representations. This disentanglement facilitates the object classifier in focusing exclusively on the semantic content, thereby enhancing its generalizability across novel target domains.

In this context, the global domain classifier, denoted as $\mathcal{F}_d$, is tasked with identifying domain identifiers based on the attended features, which are defined as $Fe_d(x) = Fe_l(x) \otimes \mathcal{A}_d + Fe_l(x)$. This is achieved through the use of a multiclass cross-entropy loss. It is worth noting that $\mathcal{F}_d$ implicitly ensures that $\mathcal{F}_G$ generates images in accordance with the specified conditioning domain identifiers. The corresponding loss function is delineated below.

$$\mathbf{L_{dom}} = \mathbb{E}_{P_{\mathcal{D}}, P_{noise}^{cs/os}} [\mathbf{L_{CE}}(\mathcal{F}_d(\underbrace{Fe_d(x_r^{rec})}_{\text{domain features}}), v_d) + \mathbf{L_{CE}}(\mathcal{F}_d(\underbrace{Fe_d(x_f^{os/cs})}_{\text{domain features}}), \hat{v}_d)]. \tag{4}$$

In contrast, the open-set classifier, $\mathcal{F}_o$, is trained to accurately identify all samples belonging to known classes while disregarding the generated pseudo-outliers by labeling them as $\mathcal{C} + 1$, from $Fe_o(x) = Fe_l(x) \otimes \mathcal{A}_o + Fe_l(x)$. This is achieved using a multiclass cross-entropy loss. Similar to $\mathcal{F}_d$, $\mathcal{F}_o$ also aids $\mathcal{F}_G$ in producing high-quality synthesized images and supplements $\mathbf{L_{rec}}$.

$$\mathbf{L_{class}} = \mathbb{E}_{P_{\mathcal{D}}, P_{noise}^{cs/os}} [\mathbf{L_{CE}}(\mathcal{F}_o(\underbrace{Fe_o(x_r)}_{\text{object features}}), \tilde{y}) + \mathbf{L_{CE}}(\mathcal{F}_o(\underbrace{Fe_o(x_f^{cs/os})}_{\text{object features}}), \tilde{y})]. \tag{5}$$

$\mathcal{F}_d$ and $\mathcal{F}_o$ work together on $Fe_l(x)$, and seek to learn the domain-specific and semantic features separately, suggesting that fact that both the networks are devoted to disentangling the latent features wisely.

---

**Algorithm 1** ODG-NET training algorithm

---

**Require:** Initialized $\mathcal{F}_G, \mathcal{F}_{im}, \mathcal{F}_v, \mathcal{F}_y, \mathcal{F}_\eta, \mathcal{F}_d, \mathcal{F}_o, \mathcal{F}_{disc}, \{\mathcal{F}_l^s\}_{l=1}^{\mathcal{S}}$
 1: **while** Not Converged **do**
 2:  Sample a batch of $(x_r, y, v_d)$ from $\mathcal{D}$ and $\eta_1 \sim \mathcal{N}(0, \mathbb{I}), \eta_2 \sim \mathcal{N}(0, \sigma), \eta_3 \sim \mathcal{N}(0, \mathbb{I})$. $\sigma$ is the noise variance for generating the pseudo-open samples.
 3:  Generate $\hat{v}_d$s and $\hat{y}$s using Dirichlet($\alpha$). $\alpha$ is the parameter of the distribution.
 4:  Obtain a batch of $x_f^{cs} = \mathcal{F}_G(x_r, \hat{v}_d, y, \eta_1)$.
 5:  Obtain a batch of $x_f^{os} = \mathcal{F}_G(x_r, v_d/\hat{v}_d, \hat{y}, \eta_2)$.
 6:  Obtain $x_r^{rec} = \mathcal{F}_G(x_f^{cs/os}, v_d, y, \eta_3)$.
 7:  Obtain $Fe_l$, the latent representation corresponding to $(x_r, x_f^{os}, x_f^{cs}, x_r^{rec})$.
 8:  Obtain $Fe_d$, the attended domain features, and $Fe_o$, the attended semantic features from $Fe_l$.
 9:  Solve: $\underset{\substack{\mathcal{F}_{im}, \mathcal{F}_v, \mathcal{F}_y, \mathcal{F}_\eta, \mathcal{F}_G, \\ \{\mathcal{F}_l^s\}_{s=1}^{\mathcal{S}}}}{\operatorname{argmin}} \underset{\mathcal{F}_{disc}}{\operatorname{argmax}} [w_{Gan}\mathbf{L_{Gan}} + w_{rec}\mathbf{L_{rec}} + w_{local}\mathbf{L_{local}}]$.
10:  Solve: $\underset{\mathcal{F}_o, \mathcal{F}_d}{\operatorname{argmin}} [\mathbf{L_{dom}} + \mathbf{L_{class}} + w_f\mathbf{L_{sem}}]$.
11: **end while**

---

**Ensuring discriminativeness of the semantic feature space**. The inherent diversity within multi-domain data poses a challenge for $\mathcal{F}_o$ in creating an optimized semantic feature space based on $\mathcal{F}_o^b$. In this space, it's expected that closed-set classes from the augmented domains should form distinct clusters, while pseudo-open-set samples should be effectively pushed away from the support region of the closed set. To enhance the discriminative qualities, we propose the utilization of a contrastive loss for closed-set samples across the augmented domain set. Simultaneously, we aim to minimize the entropy ($\mathcal{E}$) of $\mathcal{F}_o$ predictions for pseudo-open samples. Minimizing the entropy effectively acts as a weighting mechanism for $\mathcal{F}_o$ when handling pseudo-open samples. This weighting strategy increases the posterior probability $p(y = \mathcal{C} + 1|x_f^{cs/os})$ for pseudo-open samples while reducing the posteriors associated with the known-class indices $(1 - \mathcal{C})$. This approach contributes to the overall objective of creating a more discriminative semantic feature space, which is crucial for the successful separation of classes and pseudo-open samples.

To implement the contrastive loss, $\mathbf{L_{con}}$, we select an anchor sample, $x_a$, and randomly obtain a positive sample, $x_+$, which shares the same class label as $x_a$, as well as a set of negative samples, $\{x_-^m\}_{m=1}^M$, where no restrictions are imposed on the styles of the samples. The goal is to maximize the cosine similarity, $\delta$, for $(x_a, x_+)$, while minimizing it for all possible pairs of $(x_a, x_-^m)$. The semantic feature space optimization loss can be expressed as follows:

$$\mathbf{L_{sem}} = \mathop{\mathbb{E}}_{P_{\mathcal{D}}, P_{noise}^{os/cs}} [\mathcal{E}(\mathcal{F}_o(Fe_o(x_f^{os}))) + \mathbf{L_{con}}]. \tag{6}$$

where $\mathbf{L_{con}}$ is defined as follows,

$$\mathbf{L_{con}} = \left[ -\log \frac{\exp(\delta(\mathcal{F}_o^b(x_a), \mathcal{F}_o^b(x_+)))}{\sum_{m=1}^M \exp(\delta(\mathcal{F}_o^b(x_a), \mathcal{F}_o^b(x_-^m)))} \right]. \tag{7}$$

**Total loss and training**. We follow an alternate optimization strategy in each training episode for ODG-NET, mentioned in Algorithm 1. In the vanilla training stage, we train the embedding networks $(\mathcal{F}_{im}, \mathcal{F}_v, \mathcal{F}_y, \mathcal{F}_\eta)$, GAN modules $\mathcal{F}_G, \mathcal{F}_{disc}$, and the local domain-specific networks $\{\mathcal{F}_l^s\}_{s=1}^{\mathcal{S}}$, given the fixed $(\mathcal{F}_d, \mathcal{F}_o)$ to produce meaningful images. $w$s represent the loss contributions and *we set them to the value* 1 *in all our experiments.*

$$\mathop{\mathrm{argmin}}_{\substack{\mathcal{F}_{im}, \mathcal{F}_v, \mathcal{F}_y, \mathcal{F}_\eta, \mathcal{F}_G, \\ \{\mathcal{F}_l^s\}_{s=1}^{\mathcal{S}}}} \mathop{\mathrm{argmax}}_{\mathcal{F}_{disc}} [w_{Gan}\mathbf{L_{Gan}} + w_{rec}\mathbf{L_{rec}} + w_{local}\mathbf{L_{local}}]. \tag{8}$$

Subsequently, we train $\mathcal{F}_o$ and $\mathcal{F}_d$ to obtain the optimized semantic classifier keeping other parameters fixed.

$$\mathop{\arg\min}_{\mathcal{F}_o, \mathcal{F}_d} [\mathbf{L_{dom}} + \mathbf{L_{class}} + w_f\mathbf{L_{sem}}]. \tag{9}$$

**Testing**. During inference, images from $\mathcal{D}_{\mathcal{T}}$ are provided as input to $\{\mathcal{F}_l^s\}_{s=1}^{\mathcal{S}}$. The class labels with the highest softmax probability scores are predicted according to $\mathcal{F}_o$.

# 4 Experimental Evaluations

**Datasets**. We present our results on six widely used benchmark datasets for DG. Specifically, we follow the approach of Shu et al. (2021) and use the following datasets: (1) **Office-Home** Venkateswara et al. (2017), (2) **PACS** Li et al. (2017), (3) **Multi-Dataset** Shu et al. (2021). In addition, we introduce the experimental setup of ODG for two additional DG datasets, namely **VLCS** Fang et al. (2013) and **Digits-DG** Zhou et al. (2020b) in this paper. For our closed-set DG experiment, we also utilize the large-scale **DomainNet** Peng et al. (2019).

**Implementation details:** To ensure clarity, we use a ResNet-18 based backbone He et al. (2016) for $\mathcal{F}_o$ consistently, while we adopt standard architectures per benchmark for closed-DG tasks, following established literature Zhou et al. (2020b). Our attention modules $(\mathcal{A}_d, \mathcal{A}_o)$ are composed of a pair of spatial and spectral attention modules, implemented using the query-key-value processing-based approach Han et al. (2022). In total, ODG-NET comprises 48 million parameters for $\mathcal{S} = 3$, and the training stage requires 65 GFLOPS.

**Training protocol and model selection**. We employ a standardized training protocol across all datasets. During each training iteration, we first optimize Eq. 8 using the Adam optimizer Kingma & Ba (2014), with a learning rate of $2e-4$ and betas of $(0.5, 0.99)$. We then minimize Eq. 9 using Adam with a learning rate of $2e-2$ and betas of $(0.9, 0.99)$. Our batch size is typically set to 64, and we train for 30 epochs, except for DomainNet, where we use a batch size of 128 and train for 40 epochs. We follow a cross-validation approach to estimate the loss weights, holding out 10% of samples per domain and using held-out pseudo-open-set validation samples obtained through cumix Mancini et al. (2020b), that the model has not seen to select the best-performing model. In this regard, the mixup samples do not have a clear semantic meaning as they are generated by randomly combining two images. Hence, they can be considered representative open samples. We further consider $\beta = 0.5$ to put $\mathcal{W}$ as a soft constraint in $\mathbf{L_{GAN}}$. A large $\beta$ instigates the generation of ambiguous images in order to make them different from $\mathcal{D}$. Besides, we set $\alpha = 0.5$ following Shu et al. (2021).

**Evaluation protocol**. For ODG experiments, we report the top-1 accuracy for closed-set samples (Acc) and the H-score for closed and open samples. For closed-set DG experiments, we consider the top-1 performance. We report the mean $\pm$ std. over three runs.

## 4.1 Results on open DG tasks

**Baselines**. Our baseline method, AGG, involves merging the source domains with different label sets and training a unified classifier on all the classes. In comparison, we evaluate the performance of ODG-NET against traditional DG methods that are less sensitive to label changes between different source domains, as outlined in Shu et al. (2021). These include state-of-the-art meta-learning-based and augmentation-based DG methods Li et al. (2018a; 2019a); Mancini et al. (2020a); Zhou et al. (2021); Shi et al. (2021); Rame et al. (2022), heterogeneous DG Li et al. (2019b), and methods that produce discriminative and generic embedding spaces Wang et al. (2020b); Huang et al. (2020); Zhang et al. (2022). As per Shu et al. (2021), we employ a confidence-based classifier for our competitors. Here, a sample is classified as unknown if the class probabilities are below a predefined threshold. Alternatively, we also compare against the only existing ODG technique, , DAML Shu et al. (2021) and consider a variant where we combine DAML with Openmax Bendale & Boult (2016b) based OSR. Finally, we report the results of two open-set recognition baselines, Openmax Bendale & Boult (2016b) and MORGAN Pal et al. (2023b).

Table 1: Comparative analysis for PACS on ODG. (In %)

| Methods | Art | | Sketch | | Photo | | Cartoon | | Avg | |
|---|---|---|---|---|---|---|---|---|---|---|
| | Acc | H-score | Acc | H-score | Acc | H-score | Acc | H-score | Acc | H-score |
| AGG | 51.35 | 38.87 | 49.75 | 47.09 | 53.15 | 44.19 | 66.43 | 48.98 | 55.17 | 44.78 |
| OpenMax Bendale & Boult (2016b) | 53.33 | 40.83 | 55.45 | 54.18 | 73.76 | 53.29 | 73.39 | 53.87 | 63.98 | 50.54 |
| MORGAN Pal et al. (2023a) | 44.56 | 35.78 | 52.31 | 51.49 | 70.29 | 49.55 | 66.31 | 48.69 | 58.37 | 46.37 |
| MLDG Li et al. (2018a) | 44.59 | 31.54 | 51.29 | 49.91 | 62.20 | 43.35 | 71.64 | 55.20 | 57.43 | 45.00 |
| FC Li et al. (2019b) | 51.12 | 39.01 | 51.15 | 49.28 | 60.94 | 45.79 | 69.32 | 52.67 | 58.13 | 46.69 |
| Epi-FCR Li et al. (2019a) | 54.16 | 41.16 | 46.35 | 46.14 | 70.03 | 48.38 | 72.00 | 58.19 | 60.64 | 48.47 |
| PAR Wang et al. (2020b) | 52.97 | 39.21 | 53.62 | 52.00 | 51.86 | 36.53 | 67.77 | 52.05 | 56.56 | 44.95 |
| RSC Huang et al. (2020) | 50.47 | 38.43 | 50.17 | 44.59 | 67.53 | 49.82 | 67.51 | 47.35 | 58.92 | 45.05 |
| CuMix Mancini et al. (2020a) | 53.85 | 38.67 | 37.70 | 28.71 | 65.67 | 49.28 | 74.16 | 47.53 | 57.85 | 41.05 |
| Fish Shi et al. (2021) | 52.22 | 39.54 | 55.54 | 54.28 | 69.41 | 48.87 | 69.85 | 51.75 | 61.75 | 48.61 |
| Disentanglement Zhang et al. (2022) | 53.18 | 38.32 | 56.39 | 53.36 | 71.99 | 47.39 | 70.54 | 50.63 | 63.02 | 47.42 |
| Mixstyle Zhou et al. (2021) | 53.41 | 39.33 | 56.10 | 54.44 | 72.37 | 47.21 | 71.54 | 52.22 | 63.35 | 48.30 |
| DAML Shu et al. (2021) | 54.10 | 43.02 | 58.50 | 56.73 | 75.69 | 53.29 | 73.65 | 54.47 | 65.49 | 51.88 |
| DAML + OpenMax Bendale & Boult (2016a) | 52.73 | 41.28 | 57.81 | 56.82 | 74.55 | 54.55 | 75.84 | 55.96 | 65.23 | 52.15 |
| **ODG-NET** | **57.21** | **46.19** | **61.85** | **59.25** | **78.76** | **56.67** | **77.39** | **61.11** | **68.80** | **55.81** |

Table 2: Comparative analysis for Office-Home on ODG. (In %)

| Methods | Clipart | | Real-World | | Product | | Art | | Avg | |
|---|---|---|---|---|---|---|---|---|---|---|
| | Acc | H-score | Acc | H-score | Acc | H-score | Acc | H-score | Acc | H-score |
| AGG | 42.83 | 44.98 | 62.40 | 53.67 | 54.27 | 50.11 | 42.22 | 40.87 | 50.43 | 47.41 |
| OpenMax Bendale & Boult (2016b) | 43.29 | 43.67 | 62.45 | 59.86 | 56.71 | 52.29 | 48.76 | 47.54 | 52.81 | 50.84 |
| MORGAN Pal et al. (2023a) | 39.68 | 41.18 | 59.87 | 59.76 | 55.33 | 52.19 | 43.33 | 42.87 | 49.55 | 49.00 |
| MLDG Li et al. (2018a) | 41.82 | 41.26 | 62.98 | 55.84 | 56.89 | 52.25 | 42.58 | 40.97 | 51.07 | 47.58 |
| FC Li et al. (2019b) | 41.80 | 41.65 | 63.79 | 55.16 | 54.41 | 52.02 | 44.13 | 43.25 | 51.03 | 48.02 |
| Epi-FCR Li et al. (2019a) | 37.13 | 42.05 | 62.60 | 54.73 | 54.95 | 52.68 | 46.33 | 44.46 | 50.25 | 48.48 |
| PAR Wang et al. (2020b) | 41.27 | 41.77 | 65.98 | 57.60 | 55.37 | 54.13 | 42.40 | 42.62 | 51.26 | 49.03 |
| RSC Huang et al. (2020) | 38.60 | 38.39 | 60.85 | 53.73 | 54.61 | 54.66 | 44.19 | 44.77 | 49.56 | 47.89 |
| CuMix Mancini et al. (2020a) | 41.54 | 43.07 | 64.63 | 58.02 | 57.74 | 55.79 | 42.76 | 40.72 | 51.67 | 49.40 |
| Fish Shi et al. (2021) | 43.76 | 44.38 | 65.25 | 58.74 | 57.86 | 57.33 | 49.78 | 46.55 | 54.16 | 51.75 |
| Disentanglement Zhang et al. (2022) | 44.89 | 42.87 | 63.38 | 59.51 | 58.88 | 55.44 | 45.49 | 43.43 | 53.16 | 50.31 |
| Mixstyle Zhou et al. (2021) | 42.28 | 41.15 | 61.78 | 60.23 | 59.92 | 53.97 | 50.11 | 42.78 | 53.52 | 49.53 |
| DAML Shu et al. (2021) | 45.13 | 43.12 | 65.99 | 60.13 | 61.54 | 59.00 | 53.13 | 51.11 | 56.45 | 53.34 |
| DAML + OpenMax Bendale & Boult (2016a) | 45.51 | 44.25 | 60.33 | 61.46 | 60.71 | 59.67 | 51.34 | 52.34 | 54.47 | 54.43 |
| **ODG-NET** | **49.81** | **48.39** | **68.45** | **63.33** | **63.29** | **61.51** | **56.05** | **53.52** | **59.40** | **56.69** |

**Quantitative and qualitative analysis**. Tables 1-5 present a performance comparison of ODG-NET with the literature on five datasets. ODG-NET consistently outperforms others in terms of Acc and H-score for all domain combinations and the average leave-one-out case where all the domains except one are used during training and the model is validated on the held-out target domain. For example, on PACS, ODG-NET achieves an Acc of 68.80% and an H-score of 55.81%, beats the previous best of DAML+OpenMax which obtained 65.23% and 52.15%, respectively. Our method outperforms Shu et al. (2021) by $\approx 3\%$ for Office-Home and $\approx 5\%$ for VLCS and Digits-DG in H-score. For the Multi-

Table 3: Comparative analysis for VLCS on ODG. (In %)

| Methods | Caltech | | LabelMe | | Pascal VOC | | Sun | | AVG | |
|---|---|---|---|---|---|---|---|---|---|---|
| | Acc | H-score | Acc | H-score | Acc | H-Score | Acc | H-score | Acc | H-score |
| Acc | | | | | | | | | | |
| AGG | 65.49 | 62.59 | 46.15 | 42.78 | 48.29 | 44.31 | 44.48 | 40.67 | 51.10 | 47.58 |
| OpenMax Bendale & Boult (2016b) | 64.19 | 62.54 | 47.77 | 45.41 | 48.82 | 45.89 | 46.61 | 45.51 | 51.84 | 49.83 |
| MORGAN Pal et al. (2023a) | 61.59 | 59.87 | 43.33 | 40.98 | 46.71 | 40.08 | 42.22 | 41.16 | 48.46 | 45.52 |
| MLDG Li et al. (2018a) | 66.91 | 63.11 | 45.65 | 41.76 | 48.37 | 42.71 | 44.29 | 42.22 | 51.30 | 47.45 |
| FC Li et al. (2019b) | 65.59 | 60.48 | 45.23 | 44.22 | 49.23 | 45.89 | 45.32 | 44.45 | 51.34 | 48.76 |
| EPI-FCR Li et al. (2019a) | 66.81 | 62.98 | 47.83 | 45.33 | 50.22 | 45.56 | 46.03 | 44.32 | 52.72 | 49.55 |
| PAR Wang et al. (2020b) | 65.78 | 61.25 | 46.21 | 42.54 | 50.11 | 46.33 | 45.39 | 43.65 | 51.87 | 48.44 |
| RSC Huang et al. (2020) | 64.43 | 61.39 | 45.61 | 43.71 | 48.60 | 42.65 | 45.76 | 42.71 | 51.10 | 47.61 |
| CuMix Mancini et al. (2020a) | 66.21 | 63.76 | 46.72 | 45.59 | 50.54 | 45.78 | 46.38 | 45.33 | 52.46 | 50.11 |
| Fish Shi et al. (2021) | 65.82 | 62.29 | 47.66 | 46.52 | 50.11 | 45.53 | 45.54 | 43.33 | 52.28 | 49.41 |
| Disentanglement Zhang et al. (2022) | 63.27 | 61.86 | 48.65 | 45.39 | 50.53 | 43.22 | 46.72 | 45.76 | 52.29 | 49.05 |
| Mixstyle Zhou et al. (2021) | 66.11 | 63.19 | 46.72 | 46.22 | 49.75 | 46.19 | 46.62 | 46.87 | 52.30 | 50.61 |
| DAML Shu et al. (2021) | 69.18 | 64.65 | 48.22 | 47.71 | 49.87 | 47.22 | 46.87 | 46.78 | 53.53 | 51.59 |
| DAML + OpenMax Bendale & Boult (2016a) | 68.24 | 66.51 | 46.43 | 46.18 | 52.49 | 47.00 | 47.43 | 47.71 | 53.64 | 51.85 |
| **ODG-NET** | **73.42** | **69.93** | **51.89** | **51.56** | **53.44** | **52.75** | **50.21** | **50.14** | **57.24** | **56.09** |

Table 4: Comparative analysis for Digit-DG on ODG. (In %)

| Methods | MNIST | | MNIST_M | | SVHN | | SYN | | AVG | |
|---|---|---|---|---|---|---|---|---|---|---|
| | Acc | H-score | Acc | H-score | Acc | H-Score | Acc | H-score | Acc | H-score |
| AGG | 69.45 | 63.28 | 43.51 | 42.15 | 50.26 | 46.89 | 61.87 | 56.31 | 56.27 | 52.15 |
| OpenMax Bendale & Boult (2016b) | 73.87 | 65.39 | 46.71 | 44.63 | 53.87 | 48.21 | 65.55 | 61.63 | 60.00 | 54.96 |
| MORGAN Pal et al. (2023a) | 72.45 | 63.59 | 41.78 | 43.32 | 50.67 | 48.77 | 65.78 | 61.49 | 57.67 | 54.29 |
| MLDG Li et al. (2018a) | 71.33 | 69.22 | 43.19 | 41.78 | 48.73 | 45.37 | 61.28 | 58.22 | 56.13 | 53.64 |
| FC Li et al. (2019b) | 71.29 | 66.29 | 41.22 | 40.67 | 47.72 | 44.41 | 59.33 | 55.67 | 54.89 | 51.76 |
| EPI-FCR Li et al. (2019a) | 72.39 | 68.33 | 45.83 | 43.34 | 51.27 | 46.88 | 62.46 | 60.23 | 57.98 | 54.69 |
| PAR Wang et al. (2020b) | 70.88 | 67.47 | 44.62 | 42.65 | 49.34 | 45.72 | 60.23 | 57.11 | 56.26 | 53.23 |
| RSC Huang et al. (2020) | 72.77 | 66.34 | 42.27 | 41.43 | 48.32 | 45.59 | 62.41 | 57.26 | 56.44 | 52.65 |
| CuMix Mancini et al. (2020a) | 72.10 | 67.52 | 45.88 | 43.74 | 52.22 | 47.22 | 62.33 | 58.33 | 58.13 | 54.20 |
| Fish Shi et al. (2021) | 74.43 | 66.89 | 42.65 | 44.45 | 52.31 | 46.71 | 64.76 | 58.73 | 58.53 | 54.19 |
| Disentanglement Zhang et al. (2022) | 71.29 | 68.83 | 45.38 | 41.59 | 50.16 | 42.71 | 65.66 | 60.33 | 58.12 | 53.36 |
| Mixstyle Zhou et al. (2021) | 76.56 | 70.56 | 47.81 | 45.66 | 54.97 | 47.24 | 61.80 | 61.96 | 60.23 | 56.35 |
| DAML Shu et al. (2021) | 73.98 | 69.88 | 46.49 | 45.62 | 53.34 | 47.72 | 64.22 | 59.23 | 59.51 | 55.61 |
| DAML + OpenMax Bendale & Boult (2016a) | 75.77 | 71.38 | 48.51 | 47.49 | 55.61 | 49.69 | 65.49 | 62.77 | 61.34 | 57.83 |
| **ODG-NET** | **78.56** | **75.75** | **50.52** | **50.22** | **57.81** | **52.62** | **68.94** | **65.33** | **63.85** | **60.98** |

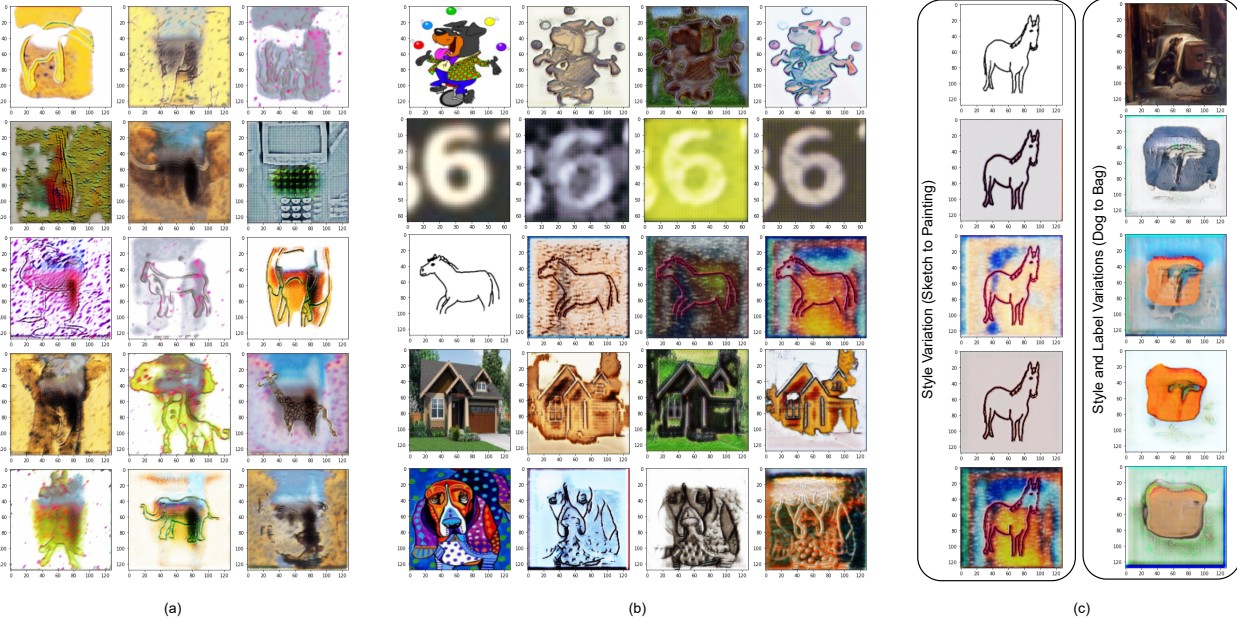

(a)  (b)  (c)

Figure 3: (a) Depiction of the generated pseudo-open-set samples by ODG-NET. (b) Depiction of the pseudo-stylized images (columns 2-4) generated by ODG-NET w.r.t. the input images mentioned in column 1. (c) Results of the intermediate images (two images for both cases.)showing the transition between a pair of domain/label.

Table 5: Comparative analysis for Multi-Dataset on ODG. (In %)

| Methods | Clipart | | Real | | Painting | | Sketch | | Avg | |
|---|---|---|---|---|---|---|---|---|---|---|
| | Acc | H-score | Acc | H-score | Acc | H-Score | Acc | H-score | Acc | H-score |
| AGG | 29.78 | 34.06 | 65.33 | 64.72 | 44.30 | 51.04 | 27.59 | 35.41 | 41.75 | 46.31 |
| OpenMax Bendale & Boult (2016b) | 36.72 | 34.41 | 63.29 | 62.88 | 44.19 | 50.75 | 34.51 | 32.29 | 44.67 | 45.08 |
| MORGAN Pal et al. (2023a) | 29.45 | 28.59 | 59.97 | 62.22 | 43.75 | 47.53 | 31.19 | 30.04 | 41.09 | 42.09 |
| MLDG Li et al. (2018a) | 29.66 | 35.11 | 65.37 | 54.40 | 44.04 | 50.53 | 26.83 | 34.57 | 41.48 | 43.65 |
| FC Li et al. (2019b) | 29.91 | 35.42 | 64.77 | 63.65 | 44.13 | 50.07 | 28.56 | 34.10 | 41.84 | 45.81 |
| Epi-FCR Li et al. (2019a) | 27.70 | 37.62 | 60.31 | 64.95 | 39.57 | 50.24 | 26.76 | 33.74 | 38.59 | 46.64 |
| PAR Wang et al. (2020b) | 29.29 | 39.99 | 64.09 | 62.59 | 42.36 | 46.37 | 30.21 | 39.96 | 41.49 | 47.23 |
| RSC Huang et al. (2020) | 27.57 | 34.98 | 60.36 | 60.02 | 37.76 | 42.21 | 26.21 | 30.44 | 37.98 | 41.91 |
| CuMix Mancini et al. (2020a) | 30.03 | 40.18 | 64.61 | 65.07 | 44.37 | 48.70 | 29.72 | 33.70 | 42.18 | 46.91 |
| Fish Shi et al. (2021) | 32.78 | 35.42 | 65.43 | 67.77 | 45.37 | 48.81 | 32.35$^{\mathcal{S}}$ | 32.45 | 43.98 | 46.11 |
| Disentanglement Zhang et al. (2022) | 28.76 | 33.33 | 64.48 | 64.44 | 42.29 | 50.05 | 30.65 | 35.87 | 41.54 | 45.92 |
| Mixstyle Zhou et al. (2021) | 30.03 | 40.18 | 64.61 | 65.07 | 44.37 | 48.70 | 29.72 | 33.70 | 42.18 | 46.91 |
| DAML Shu et al. (2021) | 37.62 | 44.27 | 66.54 | 67.80 | 47.80 | 52.93 | 34.48 | 41.82 | 46.61 | 51.71 |
| DAML + OpenMax Bendale & Boult (2016a) | 38.55 | 45.51 | 66.87 | 68.89 | 48.51 | 53.12 | 35.61 | 42.56 | 47.38 | 52.52 |
| **ODG-NET** | **40.75** | **47.54** | **69.49** | **71.22** | **50.11** | **55.39** | **37.58** | **44.10** | **49.48** | **54.56** |

dataset, ODG-NET achieves an Acc of $49.48\%$ and an H-score of $54.56\%$, which is an improvement of more than $3\%$ than Shu et al. (2021). Visually, the T-SNE Van der Maaten & Hinton (2008) Fig. 4(a) confirms the discriminative and domain-independent nature of the semantic space given the augmented source data.

Moreover, we present a collection of synthetically created images produced by our novel ODG-NET. As illustrated in Figure 3, (a) displays the generated pseudo-open-set examples, while (b) exhibits the pseudo-stylized pictures (columns 2-4) derived from their corresponding input images (column 1). This comparison highlights the evident transformation from the original input images to the synthesized pseudo images. Additionally, in (c), we demonstrate two aspects: first, the variation in artistic style of the input image, such as from sketch to painting; second, the combined shift in both style and label, exemplified by the transition from a "dog" class image to a "bag" class image.

## 4.2 Results on closed DG tasks

In the context of closed-set DG tasks, we compare the performance of ODG-NET against the existing literature, focusing on supervised pre-training methods that use meta-learning, regularization, and domain augmentation techniques Zhou et al. (2020b; 2021); Chen et al. (2021); Kang et al. (2022); Chattopadhyay et al. (2020); Xu et al. (2021); Shu et al. (2021); Shi et al. (2021); Du et al. (2020); Zhao et al. (2020), among others.

As shown in Table 6 for the five benchmark DG datasets, ODG-NET outperforms all comparative techniques in the average leave-one-out DG evaluations, despite these techniques being designed explicitly for closed-set DG. We observe an improvement of at least $3 - 4\%$ across all datasets. For DomainNet, ODG-NET achieves an average accuracy of $50.16\%$, which is $4\%$ better than the previous state-of-the-art method SWAD Cha et al. (2021), likely due to the more diversified training set on which ODG-NET is trained. Finally, in closed-set DG experiments, ODG-NET outperforms DAML Shu et al. (2021) by a significant margin of at least $5 - 7\%$.

## 4.3 Ablation analysis

**Model and loss ablation**. In Table 7, we present the effects of different components of the ODG-NET model and the loss functions for PACS and Office-Home datasets. We confirmed that embedding networks are crucial for learning latent conditioning information in a meaningful way. The model without embedding layers resulted in a performance drop of approximately $5 - 6\%$ in the H-score. Similarly, attention modules helped to highlight the style and semantic features better, and using $(\mathcal{A}_d, \mathcal{A}_o)$ resulted in a $3-4\%$ improvement in the H-score for both datasets. Additionally, we experimented with using a common backbone for the source domains instead of $\{\mathcal{F}_l^s\}_{s=1}^{\mathcal{S}}$. This approach significantly reduced the H-score by $4\%$ and $6\%$ for both datasets, indicating the importance of multi-view features learned by the domain-specific backbones.

Furthermore, $\mathcal{F}_d$ helped to discriminate the domain and semantic properties of $Fe_l(x)$, and the omission of $\mathcal{F}_d$ reduced performance by almost $5\%$. We also removed both $\mathcal{F}_d$ and the local classifiers $\{\mathcal{F}_l^s\}_{l=1}^{\mathcal{S}}$ simultaneously, resulting in a performance drop of over $6\%$. When we trained $\mathcal{F}_o$ from scratch instead of using a pre-trained ResNet-18 backbone, we observed a performance drop of around $3\%$. The pre-trained ResNet-18 backbone is already rich in discriminative information, which helps our DG tasks. Concerning the loss function of feature optimization, it is evident that both

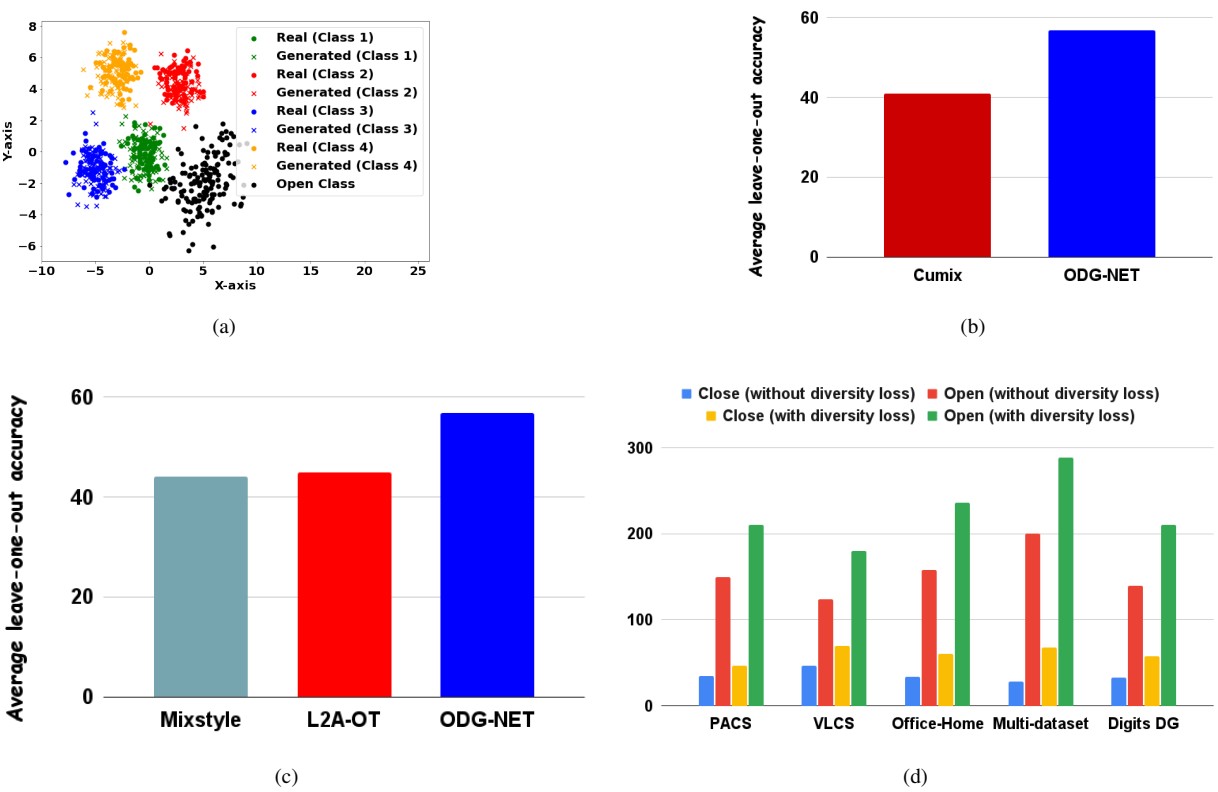

Figure 4: (a) T-SNE of real and cGAN synthesized images in the semantic feature space $\mathcal{F}_o^b$ for PACS dataset. (b) Accuracy comparison between ODG-NET and Cumix Mancini et al. (2020a) based pseudo-open sample generation. (c) Accuracy comparison between ODG-NET and Mixstyle Zhou et al. (2021) and L2A-OT Zhou et al. (2020b) based closed-set sample generation. (d) The Frećhet distance Dowson & Landau (1982) between real data and the closed and pseudo-open images generated, with and without the consideration of $\mathcal{W}$ in $\mathbf{L_{Gan}}$.

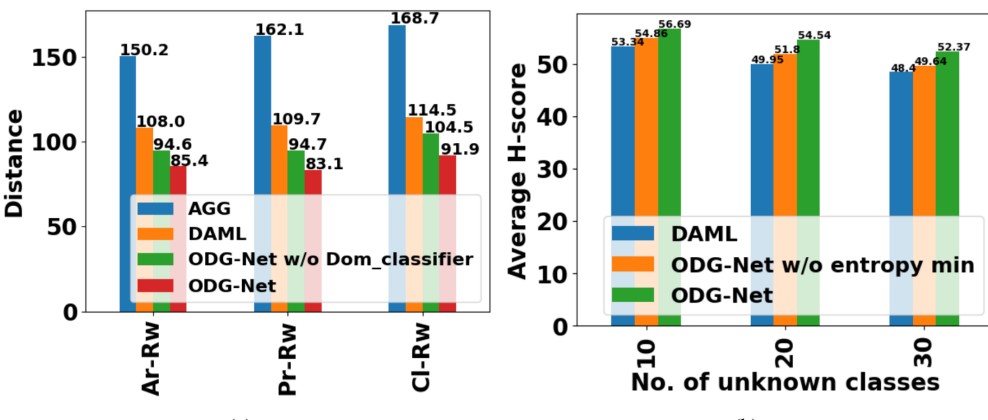

Figure 5: (a) Frećhet distance between the source and target domains for the closed-set classes. (b) Openness analysis.

Table 6: Results of PACS, VLCS, Office-Home, Digits-DG and DomainNet datasets under close-set DG. (In %)

| Methods | PACS | VLCS | Office-Home | Digits-DG | DomainNet |
|---|---|---|---|---|---|
| CCSA Motiian et al. (2017) | 79.40 | 70.20 | 64.90 | 74.50 | - |
| SFA-A Li et al. (2021c) | 81.70 | 74.00 | - | 79.60 | - |
| MetaReg Balaji et al. (2018) | 81.70 | - | - | - | 43.62 |
| MixStyle Zhou et al. (2021) | 83.70 | - | 65.50 | - | 34.0 |
| JiGen Carlucci et al. (2019) | 80.51 | 73.19 | 61.20 | 76.20 | - |
| SagNet Wu et al. (2019) | 83.25 | - | 62.34 | - | 40.30 |
| RSC Huang et al. (2020) | 85.15 | 75.43 | 63.12 | - | 38.90 |
| DDAIG Zhou et al. (2020a) | 83.10 | - | 65.50 | 77.58 | - |
| L2A-OT Zhou et al. (2020b) | 82.80 | - | 65.60 | 78.10 | - |
| FACT Xu et al. (2021) | 84.51 | - | 66.56 | 81.55 | - |
| STEAM Chen et al. (2021) | 86.60 | - | 66.80 | 83.13 | - |
| Style Neo. Kang et al. (2022) | 85.47 | - | 65.89 | - | 44.60 |
| Liu et al. Liu et al. (2021) | - | 76.48 | 67.85 | 80.02 | - |
| MMD-AAE Li et al. (2018b) | 77.00 | 72.30 | 62.70 | 74.60 | - |
| Cross-Grad Shankar et al. (2018) | 80.70 | - | 64.40 | 75.83 | - |
| MASF Dou et al. (2019) | 81.03 | 74.11 | - | - | - |
| EISNet Wang et al. (2020a) | 82.15 | 74.65 | - | - | - |
| MetaVIB Du et al. (2020) | - | 74.54 | - | - | - |
| DGER Zhao et al. (2020) | - | 74.38 | - | - | - |
| MixUp Zhang et al. (2017) | - | - | - | - | 39.20 |
| DMG Chattopadhyay et al. (2020) | - | - | - | - | 43.63 |
| SWAD Cha et al. (2021) | 88.10 | 79.10 | 70.60 | - | 46.50 |
| Fish Shi et al. (2021) | 85.50 | 77.80 | 68.60 | - | 42.70 |
| DAML Shu et al. (2021) | 82.70 | 72.95 | 67.71 | 79.89 | - |
| **ODG-NET** | **90.66** | **79.85** | **72.92** | **86.75** | **50.16** |

Table 7: Model and loss ablation analysis on PACS & Office-Home datasets. (In %)

| Model variants of ODG-NET | PACS | | Office-Home | |
|---|---|---|---|---|
| | Acc | H-score | Acc | H-score |
| - w/o $(\mathcal{F}_{im}, \mathcal{F}_v, \mathcal{F}_y, \mathcal{F}_\eta)$ | 63.43 | 50.82 | 53.07 | 50.55 |
| - w/o $\mathcal{A}_d$ and $\mathcal{A}_o$ | 66.01 | 52.93 | 56.58 | 53.53 |
| - w/o $\{\mathcal{F}_l^s\}_{s=1}^{\mathcal{S}}$, but a common backbone for the source domains | 63.73 | 51.53 | 53.14 | 50.50 |
| - w/o $\mathcal{F}_d$ | 64.98 | 52.09 | 55.55 | 52.69 |
| - w/o $\mathcal{F}_d$ and $\{\mathcal{F}_l^s\}_{s=1}^{\mathcal{S}}$ | 62.38 | 49.67 | 51.95 | 49.02 |
| - w/o Entropy loss | 66.81 | 53.15 | 57.35 | 54.36 |
| - w/o $\mathbf{L_{con}}$ | 65.36 | 51.73 | 55.40 | 51.60 |
| - w/o $\mathbf{L_{sem}}$ | 64.74 | 51.12 | 54.57 | 51.11 |
| - w/o $\mathcal{F}_d$ and $\{\mathcal{F}_l^s\}_{s=1}^{\mathcal{S}}$ and $\mathbf{L_{sem}}$ | 59.07 | 47.07 | 48.91 | 45.56 |
| - with training $\mathcal{F}_o$ from scratch | 65.96 | 52.86 | 56.38 | 53.33 |
| - w/o $\mathcal{W}$ in $\mathbf{L_{GAN}}$ | 66.26 | 53.96 | 58.38 | 54.93 |
| **Sensitivity to noise variance** **of GAN for synthesizing closed and pseudo-open samples** | | | | |
| $P_{noise}^{cs} - \mathcal{N}(0,1); P_{noise}^{os} - \mathcal{N}(0,1)$ | 65.30 | 52.51 | 55.38 | 52.53 |
| $P_{noise}^{cs} - \mathcal{N}(0,1); P_{noise}^{os} - \mathcal{N}(0,2)$ | 66.35 | 53.32 | 56.44 | 53.40 |
| $P_{noise}^{cs} - \mathcal{N}(0,1); P_{noise}^{os} - \mathcal{N}(0,3)$ | 67.29 | 54.21 | 57.18 | 54.49 |
| $P_{noise}^{cs} - \mathcal{N}(0,1); P_{noise}^{os} - \mathcal{N}(0,4)$ | 67.98 | 55.14 | 58.29 | 55.52 |
| $P_{noise}^{cs} - \mathcal{N}(0,1); P_{noise}^{os} - \mathcal{N}(0,10)$ | 68.22 | 55.37 | 58.72 | 56.11 |
| $P_{noise}^{cs} - \mathcal{N}(0,5); P_{noise}^{os} - \mathcal{N}(0,5)$ | 67.13 | 54.22 | 55.65 | 52.24 |
| **ODG-NET**$(P_{noise}^{cs} - \mathcal{N}(0,1); P_{noise}^{os} - \mathcal{N}(0,5))$ | **68.80** | **55.81** | **59.40** | **56.69** |

the closed-set contrastive and open-space entropy regularizer assists in generating a more discriminative feature space. The model without any of these losses or the full $\mathbf{L_{sem}}$ led to performance degradation by approximately 4%. Finally, we removed the diversity regularization $\mathcal{W}$ in $\mathbf{L_{GAN}}$. Although, as per Fig. 4(d), $\mathcal{W}$ induces diversity in the generated images, empirically, we observed a nominal change in the accuracy $(1-1.5\%)$ in the presence of $\mathcal{W}$.

**Comparison of our augmentation with methods from the literature**. Our augmentation technique enables more

controlled style and label mix-up, and we compared it against two types of augmentation techniques from the literature: existing style diversification approaches Zhou et al. (2020b; 2021) for closed-set classes, and Cumix Mancini et al. (2020a), which performs random image mix-up so that the generated images can be a proxy for open-space. Our results in Fig. 4(b)-4(c) demonstrate that ODG-NET performs better with our proposed augmentation. Our method is interpolation-based, which allows us to generate more style primitives than Zhou et al. (2020b). Since our method is image-based as opposed to the feature-based method of Zhou et al. (2021), we can handle the semantics better. Similarly, for pseudo-open samples, we can generate more meaningful images with varied similarities to the closed classes than the random mix-up of Mancini et al. (2020a).

**Sensitivity to variances of $P_{noise}^{os/cs}$.** In this experiment, we tune the $\sigma$ parameter of $P_{noise}^{os}$ while fixing $P_{noise}^{cs}$ for $\mathcal{F}_G$. As shown in Table 7, we observe that as we increase $\sigma$ from 1 to 5, the model performance continuously improves from $52.51\%$ to $55.81\%$ for PACS and from $52.53\%$ to $56.69\%$ for Office-Home. With high variance, the open samples are sparsely distributed, better covering the open space. However, the performance improvements are found to saturate beyond $\sigma = 5$. On the other hand, increasing the variance of $P_{noise}^{cs}$ significantly affects the performance, leading to a drop of at least $3\%$. This occurs because the generated images may deviate from the original semantic concepts, degrading the quality of the generated images.

**Frećhet distance for domain alignment**. To assess the domain independence of $\mathcal{F}_o^b$, we calculate the Frećhet distance Dowson & Landau (1982) between the closed-set classes of the source and target domains for Office-Home, with the target domain being *Real-world*. In Fig. 5(a), we show the Frećhet distance of the baseline AGG, DAML, and two variants of ODG-NET, with and without the domain classifier $\mathcal{F}_d$. The full ODG-NET produces the minimum Frećhet distance, indicating that it performs the best domain alignment among the compared models. The model without $\mathcal{F}_d$ performs poorly compared to the full ODG-NET, suggesting that the use of $\mathcal{F}_d$ helps disentangle features better, making $\mathcal{F}_o^b$ less affected by domain properties and focus on shared components.

**Sensitivity to number of target open classes**. Since there is no restriction on the number of open classes in the target domain, we are interested in assessing whether ODG-NET can handle different numbers of open classes during inference. Here, we considered the average leave-one-out H-score for Office-Home and simulated three scenarios with different numbers of open classes in the target: 10, 20, and 30. In Figure 5(b), ODG-NET consistently outperforms DAML Shu et al. (2021) by at least $3\%$ for different openness factors. The entropy minimization component of $\mathbf{L_{sem}}$ widens the gap between open and closed spaces, which is helpful in this regard. We validate this by removing the entropy component of $\mathbf{L_{sem}}$ and re-running the experiments, in which case we find that performance drops by $2 - 3\%$.

**Performance comparison when the source domains have a disjoint set of classes.** Here, we present a novel experimental scenario in ODG, where the source domains have completely different sets of classes, and the target domain consists of all the classes from the sources, plus previously unknown class samples. We compare the performance of ODG-NET with that of DAML Shu et al. (2021) for this setup in Table 8. We find that ODG-NET outperforms DAML by around $3\%$, demonstrating its robustness to extreme domain and label shifts within the source domains.

Table 8: Comparison between DAML Shu et al. (2021) and ODG-NET when the source domains have mutually disjoint classes on Office-Home dataset in terms of H-score. (In $\%$)

| Domain | Clipart | RealWorld | Product | Art | Average |
|---|---|---|---|---|---|
| DAMLShu et al. (2021) | 40.12 | 58.72 | 54.85 | 47.25 | 50.23 |
| ODG-NET | 45.69 | 62.77 | 56.95 | 49.86 | 53.81 |

# 5 Takeaways

In this paper, we present ODG-NET, a solution to the challenging problem of open domain generalization. This task combines domain generalization, open-set learning, and class imbalance in a common setting. One of the key features of ODG-NET is the novel generative augmentation, which enables continuous domain and label conditional image synthesis through interpolation of conditioning variables. This augmented training set is utilized to learn a discriminative and unbiased semantic space for an open-set classifier while minimizing the effects of domain-dependent artifacts. In our experiments, ODG-NET achieves state-of-the-art performance for both open-set and closed-set domain generalization on six benchmark datasets. We plan to extend our evaluation to more safety-critical applications in the future.

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
