# OpenReview forum: "Beyond Boundaries: A Novel Data-Augmentation Discourse for Open Domain Generalization"
_TMLR — Accepted by TMLR_

### Review · Reviewer_U3Gw · 2023-08-23

**Summary Of Contributions:**

The paper considers a novel issue  Open Domain Generalization, which aims to give open classifier to generalize well when we have multiple source domains.

Techniques:
1. a novel conditional GAN with a cycle consistency constraint
2. an anti-mode-collapse regularizer that interpolates domain and category labels

Experiments show a good performance of the proposed method ODG-NET.

**Audience:**

Yes

**Claims And Evidence:**

Yes

**Requested Changes:**

Please see the weaknesses

**Strengths And Weaknesses:**

Strengthens
1. a novel and interesting problem. (I like this problem)

2, an effective method

Weaknesses
1. Poor writing (please polish it again)
2. Please consider some baselines in OOD detection
3. I expect you to explore whether generalization can benefit detection and detection can benefit generalization.
4. Missing references:
some reference related to OOD detection, open set domain adaptation and domain generalisation, e.g.,
1. Is out-of distribution detection learnable?
2. Moderately Distributional Exploration for Domain Generalization
and so on

---

> ### Author Response · Authors · 2023-09-25
> **Author feedback for reviewer U3Gw**
>
> $$\textbf{Writing clarity.}$$
> We have re-written parts of Introduction, Related works, and Methodology in the revised manuscript.
>
> $$\textbf{Baselines in OOD detection.}$$
> In the revised manuscript, we have included results from the recent paper MORGAN [a]  and OpenMax [b] for open-set recognition.
>
> 		Methods		PACS 		Off Hm		VLCS		DgtDG		MultiDataset
> 								Acc	H	Acc	H	Acc	H	Acc	H	Acc	H
> 		Morgan		58.37	46.37	49.55	49.00	48.46	45.52	57.67	54.29	41.09	42.09
> 		OpenMax		63.98	50.54	52.81	50.84	51.84	49.83	60.00	54.96	44.67	45.08
> 		ODG-Net		68.80	55.81	59.40	56.69	57.24	56.09	63.85	60.98	49.48	54.56
>
> [a] Pal, Debabrata, et al. "MORGAN: Meta-Learning-based Few-Shot Open-Set Recognition via Generative Adversarial Network." Proceedings of the IEEE/CVF Winter Conference on Applications of Computer Vision. 2023.
>
> [b] Abhijit Bendale and Terrance E Boult. Towards open set deep networks. In Proceedings of the IEEE conference on computer vision and pattern recognition, pp. 1563–1572, 2016a.
>
> $$\textbf{Generalization helps detection and vice versa.}$$
> In the revised paper, we show a novel experiment to analyze whether generalization can benefit detection and vice versa. In particular, to check whether generalization can benefit OOD, we train a closed-set classifier given the source domains, and expect that the classifier’s entropy would be maximized for the potential target open samples. On the other hand, to analyze whether OOD can help generalization, we train the model on a single source domain, but with the open-set classification capabilities given the synthesized open samples, and we expect the model to work well on the closed-set classes from multiple target domains. The results presented below show that they do not put much effect in either of the cases.
>
> $$\textbf{Missing references.}$$
> We have added the said references in the revised manuscript.
> [a] Pal, Debabrata, et al. "MORGAN: Meta-Learning-based Few-Shot Open-Set Recognition via Generative Adversarial Network." Proceedings of the IEEE/CVF Winter Conference on Applications of Computer Vision. 2023.
> [b] Abhijit Bendale and Terrance E Boult. Towards open set deep networks. In Proceedings of the IEEE conference on
> computer vision and pattern recognition, pp. 1563–1572, 2016c
>
> $$\textbf{Is out-of distribution detection learnable?}$$
> Yes. In our case, we synthesize potential open-space samples, and learn a C+1 class classifier directly where the first C indices correspond to the closed-set classes, and the C+1th index refers to the open class samples. One of our main contributions is to learn OOD directly in this process.
>
> $$\textbf{Moderately Distributional Exploration for Domain Generalization.}$$
> Since our focus is on open domain generalization, we have discussed in detail the literature of ODG in the literature review. Nonetheless, we have briefly mentioned the closed-DG techniques as well, and also showcase how the proposed model performs in the closed-DG scenario. We have included some more references from Closed-DG in the revised manuscript.
> Here, T1= Ours w/o open samples, w/o open classifier,
> T2 = Ours w/o open samples. We show results on PACS dataset.
>
> 		Type		Art 		Sketch		Photo		cartoon		Avg
> 					Acc	H	Acc	H	Acc	H	Acc	H	Acc	H
> 		T1		59.98		60.08		77.59		75.94		68.39
> 		T2		56.69	37.77	60.75	50.67	78.11	46.89	76.69	48.76	68.03	46.02

---

### Review · Reviewer_NZiC · 2023-09-07

**Summary Of Contributions:**

This paper introduces ODG-NET, a solution for open domain generalization, combining domain generalization, open-set learning, and class imbalance.
ODG-NET employs generative augmentation for continuous domain and label conditional image synthesis, enhancing open-set classification while reducing domain-related artifacts.
It achieves state-of-the-art performance on six benchmark datasets for both open-set and closed-set domain generalization.

**Audience:**

Yes

**Claims And Evidence:**

Yes

**Requested Changes:**

As mentioned earlier, it is recommended that the authors enhance the clarity of their writing. This includes providing more comprehensive explanations for each component of the final objective and elucidating the underlying motivations, with the aim of improving the paper's overall accessibility to readers.

**Strengths And Weaknesses:**

Strengths:

1. Extensive experiments are employed to validate the effectiveness of the proposed method.

2. The proposed method achieves promising results for open and closed-set domain generalization.

Weaknesses

1. The authors should thoroughly review the writing to ensure it is easily comprehensible to a broader audience.

2. The clarity of the depiction in Figure 2 should be enhanced, and it is advisable for the authors to provide a more lucid illustration of the proposed method.

3. The proposed method appears complex due to its numerous objectives, with some not being adequately explained in the method section.

---

> ### Author Response · Authors · 2023-09-25
> **Author feedback for reviewer NZiC**
>
> 1. $$\textbf{Writing and figure clarity for broader audience}.$$
> Thank you for your kind suggestions, we have updated the write-up and figure 2 with more clarity so that it is easily understandable, and highlighted the changes with blue color.
> 2. $$\textbf{The proposed method appears complex due to its numerous objectives, with some not being adequately explained in the method section.}$$
> We propose a thorough solution for Open Domain Generalization by taking 3 aspects of learning into our solution strategy i.e., 1) Pseudo open sample generation, 2) Unbiased latent feature learning, 3) Ensuring discriminativeness, and iv) disentangling the semantic class properties from the domain artifacts from the latent features. Although it seems complex apparently, all these aspects are equally important to reach the more generalized solution, that we show through extensive experiments.

---

### Review · Reviewer_wqPp · 2023-09-12

**Summary Of Contributions:**

To address the challenge of Open Domain Generalization (ODG), this paper proposes a model called ODG-NET. The model introduces a generative augmentation framework that generates new domains of style-interpolated closed-set images and new pseudo-open images by interpolating the content of paired training images. It also addresses the problem of style bias by representing all images in terms of all source domain properties. As a result, a multi-class semantic object classifier incorporating both closed and open class classification features is trained, along with a style classifier to identify style primitives.  Experimental results on six benchmark datasets show that ODG-NET outperforms the baseline method.

**Audience:**

Yes

**Claims And Evidence:**

Yes

**Requested Changes:**

- The text in Figure 2 is too small and should be made easier to read. Also, it would be better to describe the function of each module in the figure.

- In Figure 2, \mathcal{F}^b_{1S} should be \mathcal{F}^b_{l1} and \mathcal{F}^c_{1S} should be \mathcal{F}^c_{l1}.

- Right brackets are missing in the denominator and numerator of equation (7).

**Strengths And Weaknesses:**

Strength

- This paper proposes a new network structure called ODG-NET to address the Open Domain Generalization problem.

- ODG-NET constructs an open-set classifier by acquiring discriminative, unbiased, and disentangled semantic embedding spaces.

- To synthesize diverse augmented images from the source domain, this paper proposes a novel conditional GAN with cycle consistency constraints to interpolate domain and category labels and regularization to avoid modal collapses.

- Experimental results on six benchmark datasets show that ODG-NET outperforms the baseline method.

Weaknessess

- The network as a whole is admittedly new. On the other hand, many of the individual methods are techniques that have already been used for domain generalization and domain adaptation, and have marginal novelty.

- The network structure is complex in order to incorporate different properties.

- Similarly, many loss functions are combined, and the methods lack simplicity.

- Many hyperparameters, such as \alpha, \beta, \epsilon, and different weights w for loss functions, which are difficult to tune.

- Large performance variations depending on the noise parameters.

---

> ### Author Response · Authors · 2023-09-25
> **Author feedback for reviewer wqPp**
>
> 1. $$\textbf{Marginal novelty.}$$
> We humbly differ from your opinion. Given ODG is a new problem, and the existing works are based on a confidence based classification which are affected by the confidence misclassification. In opposition, we propose a new direction to solve this problem by directly training a unified closed-open set classifier, where we propose a novel generative way for hallucination pseudo-open samples. Besides, we systematically consider all the critical aspects of solving DG, unbiased, discriminative, and disentangled latent feature learning. This makes the whole model apparently complex, but these aspects, taken together, produce improved performance for both open and closed classes. To the best of knowledge, all these aspects are used together judiciously for aDG setup for the first time.
> 2. $$\textbf{Complex network structure and loss terms.}$$
> We present a comprehensive solution for Open Domain Generalization (ODG) that addresses three fundamental aspects of learning:
> $$\textbf{Pseudo Open Sample Generation:}$$ We tackle the challenge of generating representative pseudo-open samples, a critical component in training models to generalize effectively across diverse domains.
> $$\textbf{Unbiased Latent Feature Learning:}$$ Our approach ensures that the latent features extracted are unbiased and free from domain-specific artifacts, enhancing the model's adaptability.
> $$\textbf{Discriminative Feature Space:}$$ We emphasize the importance of creating a feature space that inherently discriminates among classes, enabling the classifier to make precise distinctions across various categories.
> Although it seems complex apparently, all these aspects are equally important to reach the more generalized solution that we show through extensive experiments.
> 3. $$\textbf{Hyper-parameter tuning.}$$
> We performed the standard cross-validation approach for obtaining the optimal hyper-parameters, which is standard in the DG literature, and we follow the same.
> 4. $$\textbf{Large performance variations depending on the noise parameters.}$$
> In Table 5, we present the ACC (Accuracy) and H-score results for two datasets. Notably, we observe that both metrics exhibit variations within the range of 3-4%. These variations stem from the generation of open samples, where increased noise variations lead to the creation of diverse image types, consequently impacting the performance of open-class classification. This trend becomes evident as the results saturate with higher noise levels.
> Our findings highlight a consistent pattern across all datasets: optimal results are achieved when closed samples are generated with a low noise variance (1), while open samples benefit from a moderately large noise variance (5). This setting provides the most effective performance for the tasks at hand.
> 5. $$\textbf{Requested Changes.}$$
> We update the figure 2 along with the description of each module and equation 7 in the revised manuscript.

---

### Decision · Action_Editor_DZuz · 2023-10-30

**Recommendation:** Accept with minor revision

**Comment:**

Reviewers requested further polishing and improvement of the text. Moreover, the paper needs some polishing also with respect to figure and table spacing and sizes - it seems that the paper was compressed in that regard to fit the 12 page limit. Please:
- Make sure that there is enough space around figures (e.g. Fig2 would be better on the top of the page, with extra margins,  Fig 4 is also hard to read, enlarging the figures and placing them in a 2 by 2 grid would be better, etc)
- Make sure that the fonts in the tables are not much smaller than actual text - tables are now very hard to read.  Tab1 and Tab2 can be split into two each, ie one per dataset, Tab3,4 and 5 can cover more width so that they are readable. Also please use \citep{} for all paper references in the Tables (ie so that the reference is in a parenthesis)

It is ok of the paper ends up being slightly longer than 12 pages, as long as it is readable.

**Audience:**

The paper deals with the task of Open Domain Generalization which is a more realistic setup for domain generalization (DG). The DG community would find the method and results of this paper interesting.

**Claims And Evidence:**

The paper makes a number of claims/contributions for the task of Open domain Generalization all of them sufficiently supported.

---

> ### Author Response · Authors · 2023-11-26
> **Author Feedback to Action Editor**
>
> We thank you for your kind reviews and we have adopted the changes as suggested by you in our final camera ready submission.